# Optimizing Surface Characteristics of Stainless Steel (SUS) for Enhanced Adhesion in Heterojunction Bilayer SUS/Polyamide 66 Composites

**DOI:** 10.3390/polym16192737

**Published:** 2024-09-27

**Authors:** Sang-Seok Yun, Wanjun Yoon, Keon-Soo Jang

**Affiliations:** 1Department of Polymer Engineering, School of Chemical and Materials Engineering, The University of Suwon, Hwaseong-si 18323, Republic of Korea; 2CS Innovation Co., Hwaseong-si 18559, Republic of Korea

**Keywords:** SUS, polyamide, heterojunction bilayer composite, surface treatment, adhesion

## Abstract

The increasing environmental concerns and stringent regulations targeting emissions and energy efficiency necessitate innovative material solutions that not only comply with these standards but also enhance performance and sustainability. This study investigates the potential of heterojunction bilayer composites comprising stainless steel (SUS) and polyamide 66 (PA66), aiming to improve fuel efficiency and reduce harmful emissions by achieving lightweight materials. Joining a polymer to SUS is challenging due to the differing physical and chemical properties of each material. To address this, various surface treatment techniques such as blasting, plasma, annealing, and etching were systematically studied to determine their effects on the microstructural, chemical, and mechanical properties of the SUS surface, thereby identifying mechanisms that improve adhesion. Chemical etching using HNO_3_/HCl and CuSO_4_/HCl increased surface roughness and mechanical properties, but these properties decreased after annealing. In contrast, K_3_Fe(CN)_6_/NaOH treatment increased the lap shear strength after annealing. Blasting increased surface roughness and toughness with increasing spray pressure and further enhanced these properties after annealing. Contact angle measurements indicated that the hydrophilicity of the SUS surface improved with surface treatment and further improved due to microstructure formation after annealing. This study demonstrates that customized surface treatments can significantly enhance the interfacial adhesion and mechanical properties of SUS/polymer heterojunction bilayer composites, and further research is recommended to explore the long-term stability and durability of these treatments under various environmental conditions.

## 1. Introduction

Environmental concerns and stringent regulations targeting emissions and energy efficiency require innovation with materials that not only adhere to these standards but also enhance performance and sustainability [1,2,3]. As environmental regulations are gradually tightening worldwide owing to global warming, air pollution, climate change, and the increased CO_2_ emissions from fossil fuel energy, various research has been conducted to replace existing metal materials with the main goals of improving fuel efficiency and reducing harmful gases [4,5,6]. Research aimed at improving fuel efficiency can be broadly categorized into four main approaches: first, increasing the efficiency of power systems, such as engines [7,8,9]; second, reducing air resistance through design improvements [10,11,12]; third, using alternative energies such as electricity and hydrogen [13,14]; and, finally, lightweighting materials [15,16,17,18,19,20,21,22]. Lightweighting is particularly favored because of its advantages, including low production costs and the ability to meet the social demand for short cycles of vehicle model changes and product improvements. In addition, it can be applied to a wide range of areas, including vehicle interior and exterior parts, functional components of the engine room, electronic systems, fuel systems, airbags and safety belts, drive components, and chassis [17,21,22,23,24,25,26,27,28]. However, the replacement of traditional metals with polymers in automobile applications presents substantial challenges, notably in terms of safety [29,30,31]. Metals, especially stainless steel (SUS), are traditionally favored for their high strength and durability, which are essential for ensuring safety in critical structural and functional components [32,33]. Polymers, while advantageous for their light weight and corrosion resistance, often do not meet the identical safety standards without significant engineering and material enhancement [34,35]. Thus, the integration of SUS and polymers into a heterojunction bilayer composite can reduce the weight while maintaining the mechanical properties for the safety [32,36].

The most crucial factor in these lightweight heterojunction bilayer composite materials is the adhesion between metals and polymers. However, the difference in the physical and chemical properties of metals and polymers, such as disparity in thermal expansion coefficients and surface energy, and residual stress at the material interfaces, often leads to weak interfacial bonding, which can undermine the structural integrity and functional reliability of the composites. To address these issues, interfacial compatibilizers and surface treatment methods can be utilized [32,37,38]. To enhance the interfacial adhesion between SUS and polymers in heterojunction bilayer composites, the use of interfacial compatibilizers presents a promising strategy. Interfacial compatibilizers, such as coupling agents, block copolymers, and grafted polymers, act as molecular bridges that can modify the surface energy and reduce the interfacial tension between dissimilar materials [39,40,41]. These agents work by introducing functional groups that can chemically or physically bond with both the metal and polymer phases, thereby promoting better adhesion at the interface. For instance, silane coupling agents have been shown to effectively enhance adhesion between metal oxides and polymers by forming covalent bonds with hydroxyl groups on metal surfaces, while their organofunctional groups interact with the polymer matrix [42]. In addition, maleic anhydride-grafted polymers and block copolymers can create entanglements and interpenetrating networks, further strengthening the interfacial region [43,44]. By using interfacial compatibilizers, it is possible to address the inherent differences in thermal expansion coefficients and surface energies between metals and polymers, mitigating residual stresses and potential delamination under mechanical or thermal loading. Integrating these compatibilizers into the surface treatment processes explored in this study could offer an additional approach to optimize the adhesion and mechanical performance of SUS/polymer composites, paving the way for their broader application in lightweight and high-performance materials.

Surface treatments are critical in enhancing the interfacial adhesion between SUS and polymers in heterojunction bilayer composites, modifying the surface properties of the metal to improve compatibility with the polymer matrix. These treatments, which include mechanical, chemical, and plasma-based techniques, aim to increase surface roughness, introduce functional groups, and alter surface energy to facilitate better bonding. Mechanical treatments such as sandblasting or abrasive blasting create a roughened surface that increases the surface area available for bonding and helps achieve a stronger mechanical interlock with the polymer [45,46]. Chemical treatments, including acid etching with solutions like HNO_3_/HCl or CuSO_4_/HCl, can further modify the surface by removing contaminants and introducing polar groups, which enhance the chemical affinity between the metal and polymer. Plasma treatments, particularly oxygen plasma, are effective in introducing oxygen-containing functional groups (e.g., hydroxyl, carboxyl) that increase the surface energy and improve wettability, promoting stronger interfacial adhesion [47,48]. Each surface treatment method offers unique advantages, and their effects can be further optimized by combining them with thermal treatments, such as annealing, which can relieve internal stresses and promote additional chemical bonding. Thus, employing these surface treatment techniques in conjunction with interfacial compatibilizers provides a multifaceted approach to maximizing the adhesion and overall performance of SUS/polymer composites, crucial for their use in demanding applications where mechanical strength and durability are essential.

Addressing this challenge, the current study focuses on the application of various surface treatment techniques—blasting, plasma, annealing, and acid treatments—specifically tailored to modify the surface characteristics of SUS [32,49,50,51]. By systematically investigating the effects of each treatment and combination on the microstructural, chemical, and mechanical properties of SUS surfaces, this research aims to elucidate the mechanisms by which surface modifications contribute to improved adhesion and to identify the most effective treatment protocols.

To sum up, the objective and novelty of this research lie in its comprehensive approach to addressing the critical challenge of interfacial bonding between SUS and polymers. By exploring a diverse range of surface treatments and their effects when combined with annealing, this study provides new insights into optimizing surface properties to achieve robust adhesion in metal/polymer composites. Unlike previous studies, which often focused on a single treatment method, this work systematically compares multiple techniques and their impact on both the surface characteristics and mechanical performance of the bilayer composites. The findings offer a potential pathway for developing lightweight, high-performance materials that meet the increasingly stringent environmental and safety standards in various applications, particularly in the automotive and aerospace industries.

## 2. Experimental

### 2.1. Materials

The reagents used in this study, including hydrochloric acid (36% HCl in aqueous solution), sulfuric acid (95% H_2_SO_4_ in aqueous solution), sodium hydroxide (98% NaOH in pellet form), copper sulfate pentahydrate (99% CuSO_4_·5H_2_O in powder), and potassium ferricyanide (98% K_3_Fe(CN)_6_ in powder), were supplied by Samjeon Pure Chemical Industries Ltd. (Pyeongtaek-si, Republic of Korea). Nitric acid (60% HNO_3_ in aqueous solution) was provided by Daejung Chemicals & Metals Co., Ltd. (Siheung-si, Republic of Korea), while the stainless steel (SUS 304) and PA66 (PA66 2500K) were sourced from Anhyeon Stainless (Jongno-gu, Seoul-si, Republic of Korea) and Shenma Industrial Co. (Pingdingshan City, China), respectively.

### 2.2. SUS Surface Treatment Methods

The surface of SUS was modified using both mechanical and chemical etching methods. For chemical etching, a commonly used SUS etchant was employed as detailed in Table 1 [32,52,53,54]. The etchant was applied at temperatures of 30 °C and 50 °C, with treatment times of 1, 3, and 5 min, to reveal differences in surface morphology. Dry sandblasting was performed using aluminum oxide as an abrasive particle, with particle sizes of 36, 89, and 560 μm and blasting pressures of 2, 4, and 7 psi, to examine the effects on adhesion due to pressure variations, as shown in Table 2. In addition, oxygen plasma was utilized under separate conditions, where plasma particles reacted with the surface atoms of the material, forming oxygen functional groups such as carbonyl (C=O), carboxyl (-COOH), and hydroxyl (-OH) groups, chemically modifying the surface and increasing its hydrophilicity. This was compared with the previous etching methods. The plasma treatment was performed using a Compact Plasma Cleaner (PDC-32-G-LD, MTI Co., Republic of Korea). Oxygen (100 mL/min) and argon (100 mL/min) were simultaneously injected into a chamber (diameter 3 in, length 6.5 in). RF power, voltage, and current were set at 18 W, 720 V, and 25 mA, respectively, with treatment lasting for 10 min. Furthermore, SUS specimens treated under each etching condition and with oxygen plasma were annealed in an electric furnace at 800 °C for 2 h to compare changes in adhesion before and after microstructural changes.

The joining process involved positioning the treated SUS specimen on a hot plate preheated to 260 °C. The dimensions of the SUS were 8.5 cm × 2.8 cm (23.8 cm^2^), whereas PA66 was placed according to ISO 527-2 type 1A dimensions. As depicted in Appendix A, a PA66 sheet with dimensions of 1.9 cm × 0.7 cm (1.33 cm^2^) was placed on the SUS. To ensure uniform contact, a 200 g weight was applied over the designated bonding area, and the assembly was heated for 7 min to facilitate the bonding process.

### 2.3. Characterization

#### 2.3.1. Scanning Electron Microscopy

Scanning electron microscopy (SEM; Apro, FEI Co., Hillsboro, OR, USA) was utilized to observe the surface morphologies of both untreated and surface-treated SUS specimens, with an electron beam voltage of 10 kV and a magnification of ×5000. Specimens for SEM measurements were obtained from lap shear strength tests. For SEM analysis, a 5–10 nm thick gold layer was coated on the SUS specimens using a sputter coater (Cressington 108 Auto Sputter Coater, Ted Pella Inc., Redding, CA, USA).

#### 2.3.2. Single Lap Shear Test

A universal testing machine (UTM; VT-1T, Vluchem IND., Seoul, Republic of Korea) according to ISO 527-2 type 1A was utilized to carry out lap shear tests. The cross-section of samples had dimensions of 19 mm × 7 mm and the gauge length was 245 mm. The cell load capacity was set to 10 kN. Specimens were elongated at a constant crosshead speed of 48 mm/min at room temperature (22–24 °C). The mean values of lap shear strengths were calculated using five specimens. Toughness was quantified as the area enclosed by the stress–strain (S–S) curve and the X-axis.

#### 2.3.3. Contact Angle Test

A contact angle tester (Phoenix-MT, SEO Co., Suwon-si, Republic of Korea) was employed to measure contact angles between the SUS and deionized water (pH: 5.7, conductivity: 0.0019 mS/cm). Ten contact images were acquired for each specimen at room temperature with a camera capture speed of 1 fps, zoom lens of ×12, and an LED light source. The accuracy of the static contact angle measurement was 0.1°. The specimen dimensions were 19 mm × 7 mm.

#### 2.3.4. Atomic Force Microscopy

Atomic force microscopy (AFM, NX10, Park Systems Co., Suwon-si, Republic of Korea) was used to observe the SUS surfaces, using a non-contact mode. Specimens had dimensions of 1 mm × 1 mm with scan settings of 10 μm × 10 μm and a rate of 1.0 Hz.

## 3. Results and Discussion

The surface treatments for metals influence the surface roughness, morphology, functionality, and hydrophilicity, thereby affecting the bonding strength of heterojunction bilayer metal/polymer composites. Figure 1 illustrates the comparison of surface morphology before and after annealing for pristine and plasma-treated SUS without polymer layers. Both pristine and plasma-treated SUS without annealing exhibited smooth surfaces with minimum roughness, whereas the annealed surfaces revealed the formation of porous microstructures, indicating significant morphological changes due to the plasma treatment. Annealing can affect the microstructure of SUS through recrystallization, which can lead to changes in porosity and surface roughness. The exact effects depend on the initial surface condition, the specific annealing parameters (temperature, time, and cooling rate), and the chemical composition of the SUS. Properly controlled annealing can help reduce surface defects, smooth rough surfaces, and optimize the material properties for specific applications. However, improper annealing conditions can have adverse effects, such as increased porosity or undesirable surface roughness.

Appendix A provide detailed SEM images of SUS surfaces treated with different etching acids under various etching conditions, including time and temperature variations, both before (Appendix A) and after (Appendix A) annealing. The surface roughness of SUS increased as a function of processing time and temperature across all etching acids. In particular, the HNO_3_/HCl-treated surfaces appeared bulkier, whereas the CuSO_4_/HCl-treated surfaces exhibited a more porous microstructure, suggesting a different etching mechanism leading to increased surface area. The surface roughness of SUS treated with K_3_Fe(CN)_6_/NaOH before annealing showed similar roughness to pristine SUS, implying minimal surface modification. K_3_Fe(CN)_6_/NaOH-treated surfaces, before annealing, showed similar roughness to pristine SUS, implying minimal surface modification. Post annealing, as shown in Appendix A, all samples exhibited finer structures with micropores, suggesting that annealing facilitates the development of a microporous surface morphology which can enhance adhesion properties.

Figure 2 compares surface morphologies of SUS treated with different blasting abrasive sizes and pressures before annealing. The surface became more porous with increasing abrasive size, indicating that larger abrasives removed more materials and created deeper indentations. Variation in blasting pressure had a negligible effect on the surface morphology. Figure 3 presents SEM images of SUS treated with different blasting abrasive sizes and pressures after annealing. The sandblasting after annealing showed the most effective surface treatment, creating a significantly roughened and porous surface. This enhanced roughness after annealing was likely to improve the bonding strength of the heterojunction bilayer composites. Overall, the study demonstrates that both chemical etching and mechanical blasting treatments, especially when combined with annealing, are effective in modifying the surface characteristics of SUS. These modifications are crucial for enhancing the adhesion in metal/polymer bilayer composites, thereby potentially improving their performance in practical applications.

To confirm the surface morphologies of SUS under different treatments and conditions, AFM was conducted, as displayed in Figure 4, Figure 5, Figure 6 and Appendix A. Figure 7, Figure 8 and Figure 9 highlight the mean surface roughness measured by AFM under different conditions for each treatment method. The mean surface roughness of SUS was significantly influenced by various etching methods and conditions. Figure 7 shows that the mean surface roughness for pristine and plasma-treated SUS before annealing was 279 nm and 224 nm, respectively, which increased to 306 nm and 262 nm after annealing, indicating that annealing generally increased the surface roughness. Figure 8 illustrates the surface roughness under various etching conditions and temperatures before and after annealing. SUS surfaces treated at 50 °C generally exhibited higher mean surface roughness, compared to those treated at 30 °C. The surface roughness of SUS increased with increasing etching time across all samples. For instance, the mean surface roughness of SUS treated with HNO_3_/HCl increased from 279 nm to 513 nm at 30 °C and from 279 nm to 659 nm at 50 °C with increasing etching time. The surface etching treatment of CuSO_4_/HCl increased the mean surface roughness from 279 nm to 437 nm at 30 °C and from 279 nm to 506 nm at 50 °C as a function of etching time. Annealing introduced variability in surface roughness, making it challenging to predict the outcomes. Thus, to further examine the annealing effect, it is essential to conduct additional tests on the contact angle of SUS and mechanical properties of the heterojunction bilayer SUS/PA66 composites. Figure 9 presents the mean surface roughness of SUS treated by sandblasting with different blasting abrasive sizes and pressures. The surface roughness of SUS increased with increasing blasting pressure, although an insignificant change was observed in SEM images. Using large abrasive particles before annealing resulted in a high mean surface roughness ranging from 588 nm at 2 psi to 1313 nm at 7 psi. After blasting, annealing reduced the mean surface roughness for SUS treated with large blasting abrasives across all pressures.

Figure 10a, Appendix A show the stress–strain curves of heterojunction bilayer SUS/PA66 composites subjected to different treatment methods and conditions, both before and after annealing. Figure 10b illustrates the lap shear strength of heterojunction bilayer SUS/PA66 composites using pristine and plasma-treated SUS, also before and after annealing. The plasma treatment for SUS surfaces increased the lap shear strength of composites. This increase was attributed to the formation of hydroxyl groups on the surface due to the plasma treatment. Annealing rarely influenced the lap shear strength for the bilayer composites, whether using pristine or plasma-treated SUS, because the annealing process removed the functional groups generated by plasma treatment. Unlike other samples, the K_3_Fe(CN)_6_/NaOH-treated sample showed that 1 min of treatment resulted in higher elongation at break but lower strength compared to the pristine sample. The 3 min treatment reduced the elongation at break and increased the strength, as shown in Appendix A. This indicates that the 1 min treatment may be slightly crosslinked with the surfaces with physical and chemical interactions. However, after 3 min of treatment, the interactions may become robust with tighter network structures.

Figure 11 and Figure 12 display the lap shear strength of the composites treated with different etching acids and conditions (time and temperature) before (Figure 11) and after (Figure 12) annealing. The lap shear strength increased with increasing etching time and temperature. However, annealing also reduced the lap shear strength. Similar to in the case of plasma treatment, this is because the annealing process removed the functional groups generated by the etching acids. For instance, the lap shear strength of the composites using SUS treated with CuSO_4_/HCl increased with increasing treatment time at 30 °C from 2.32 MPa to 5.53 MPa, and at 50 °C from 2.32 MPa to 7.19 MPa. After annealing, the shear strength values decreased to 2.77 MPa at 30 °C and 4.21 MPa at 50 °C. In contrast, the lap shear strength of composites using SUS treated with K_3_Fe(CN)_6_/NaOH increased after annealing because the etching effect was negligible for SUS, whereas the annealing effect was more pronounced. Figure 13 displays lap shear strength values of the composites using SUS treated by sandblasting with different blasting abrasive sizes and pressures. The lap shear strength increased with increasing blasting pressure. However, the abrasive size and annealing barely influenced the lap shear strength. For example, the lap shear strength of the composites with an abrasive size of 560 µm increased from 2.19 MPa at 2 psi to 4.48 MPa at 7 psi.

Similar to the lap shear strength results, the plasma treatment for SUS surfaces increased the toughness of composites as shown in Figure 14. This increase was attributed to the formation of hydroxyl groups on the surface due to the plasma treatment. Annealing rarely influenced the mechanical properties for the bilayer composites, whether using pristine or plasma-treated SUS. Figure 15 displays the toughness of heterojunction bilayer SUS/PA66 composites using SUS treated with different etching acid solutions, such as HNO_3_/HCl, CuSO_4_/HCl, and K_3_Fe(CN)_6_/NaOH. The toughness for all samples treated at 50 °C was higher than that of samples treated at 30 °C. The toughness for all samples regardless of the type of etching solution increased with increasing etching time. For instance, the toughness for SUS treated with HNO_3_/HCl at 30 °C and 50 °C increased from 1.1 J/m^2^ to 4.4 and 7.4 J/m^2^, respectively, as a function of etching time. However, annealing resulted in a reduction in the toughness for all samples treated with HNO_3_/HCl and CuSO_4_/HCl under identical conditions. This suggests that annealing may remove the functional groups generated by the etching process, thereby diminishing the enhanced properties imparted by these treatments. In contrast, the toughness for the samples treated with K_3_Fe(CN)_6_/NaOH was slightly increased owing to annealing, which suggests that the etching of K_3_Fe(CN)_6_/NaOH on functionality and surface morphology was less effective than the annealing.

Figure 16 presents the toughness values of heterojunction bilayer SUS/PA66 composites subjected to different blasting conditions before and after annealing. The toughness increased as a function of blasting pressure, regardless of blasting abrasive size. For example, the toughness of the composites using SUS treated by sandblasting with a blasting abrasive size of 36 µm increased up to 2.75 J/m^2^ after annealing. The toughness is closely related to both surface roughness and morphology. While the blasting abrasive size influenced the surface roughness and morphology, no clear trend was observed regarding the effect of abrasive size on toughness. The annealing process tended to enhance the toughness of the composites using SUS treated by sandblasting, particularly at higher blasting pressures. This enhancement could be attributed to changes in the microstructure and the relief of internal stresses induced by the mechanical etching and subsequent thermal treatment. Overall, the results highlight the complex interaction between mechanical surface modification, material response to heat treatment, and the resulting mechanical properties. Understanding this interplay is crucial for optimizing the performance of heterojunction bilayer composites.

The contact angles of surfaces treated under various etching conditions were measured using deionized water (DI water). The contact angle images are presented in Figure 17, Figure 18, Figure 19 and Appendix A. The graphs showing changes in contact angles over time are compiled in Figure 20, Appendix A. Figure 17 shows the contact angle images of pristine and plasma-treated SUS before and after annealing. The contact angle of pristine SUS was approximately 80°, typical for SUS surfaces. After annealing, as shown in the SEM images in Figure 1, the formation of microstructures on the surface led to a decrease in the contact angle. The surface morphology of SUS treated by plasma appeared to be similar to that of the pristine SUS. However, the formation of hydroxyl moieties caused by the plasma treatment resulted in a reduced contact angle because of interactions between hydroxyl groups and water. After annealing, the contact angle further decreased due to the combined effects of hydroxyl groups and the formation of microstructures during the annealing process. Appendix A show the contact angle images of SUS treated with etching acids under various etching conditions before and after annealing, respectively. For example, the contact angle of SUS treated by HNO_3_/HCl at 30 and 50 °C decreased from 77° to 61° and 55°, respectively, with increasing treatment time. After annealing, the contact angle further decreased from 61° to 44° at 30 °C, and from 55° to 20° at 50 °C. Figure 18 and Figure 19 show the contact angle images of SUS treated by sandblasting with various blasting abrasive sizes and pressures before and after annealing, respectively. The contact angle of SUS treated with a blasting abrasive size of 560 µm decreased to 43.0° as the blasting pressure increased up to 7 psi. Annealing further decreased the contact angle because of similar reasons to those discussed above.

## 4. Conclusions

This study explored various surface treatments, including plasma treatment, chemical etching, and sandblasting, to enhance the adhesion and mechanical properties of SUS/polymer heterojunction bilayer composites. The findings demonstrate three main properties: (1) Surface Morphology and Roughness: Plasma treatment and chemical etching significantly increased the surface roughness of SUS, with annealing further influencing these effects by altering microstructures. Sandblasting also enhanced roughness, particularly with larger abrasive sizes and higher pressures. (2) Mechanical Properties: Plasma treatment improved lap shear strength and toughness due to hydroxyl group formation, while annealing generally reduced these benefits by removing functional groups. Chemical etching with HNO_3_/HCl and CuSO_4_/HCl increased mechanical properties, but these were diminished by annealing. In contrast, K_3_Fe(CN)_6_/NaOH-treated composites showed increased shear strength after annealing. (3) Hydrophilicity and Toughness: Plasma treatment and chemical etching improved the hydrophilicity of SUS surfaces, which was further enhanced by annealing. Toughness generally increased with etching time and temperature, while annealing showed varied effects depending on the treatment type. These results highlight the complex interactions between surface treatments and annealing, providing insights for optimizing SUS/polymer composites for lightweight and high-performance applications. Further research is recommended to examine the long-term durability of these treatments under varying conditions.

## Figures and Tables

**Figure 1 polymers-16-02737-f001:**
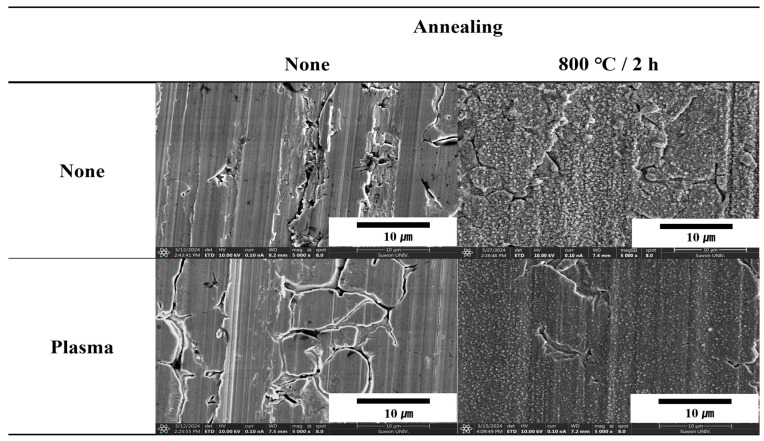
SEM images of pristine and plasma-treated SUS surfaces before and after annealing.

**Figure 2 polymers-16-02737-f002:**
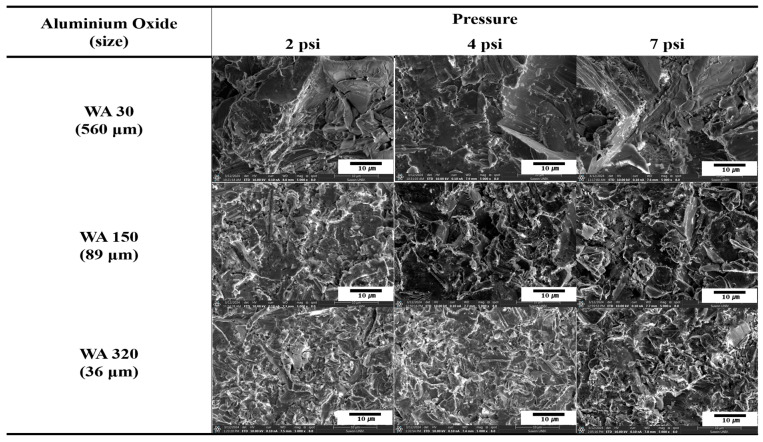
SEM images of SUS treated by blasting with various abrasive sizes and blasting pressures before annealing.

**Figure 3 polymers-16-02737-f003:**
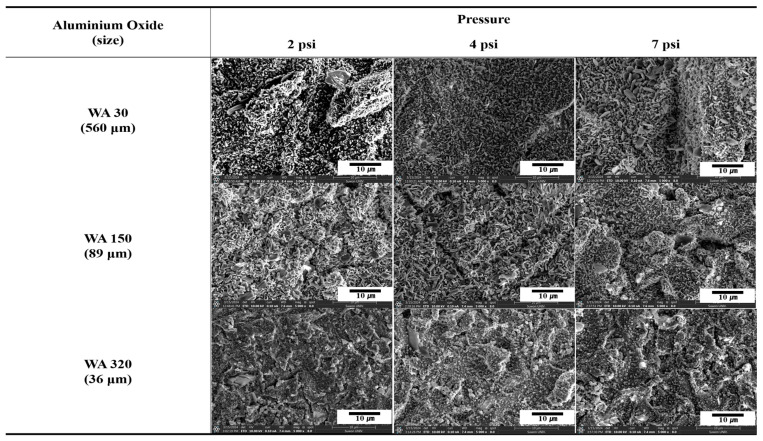
SEM images of SUS treated by blasting with various abrasive sizes and blasting pressures after annealing.

**Figure 4 polymers-16-02737-f004:**
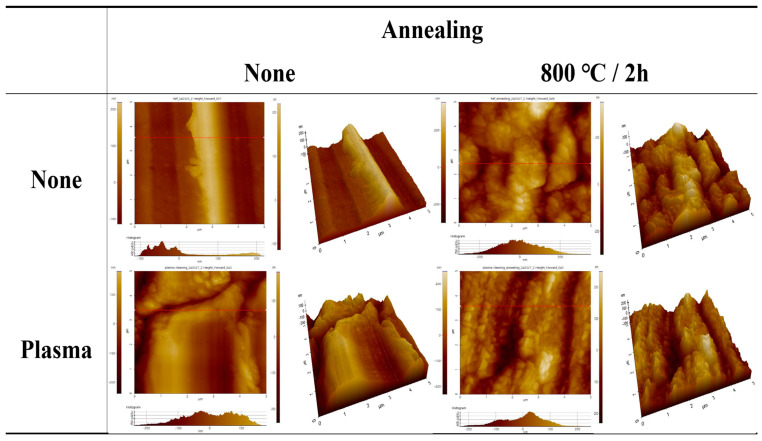
AFM images of pristine and plasma-treated SUS before and after annealing. The red line indicates the position of 3D examination.

**Figure 5 polymers-16-02737-f005:**
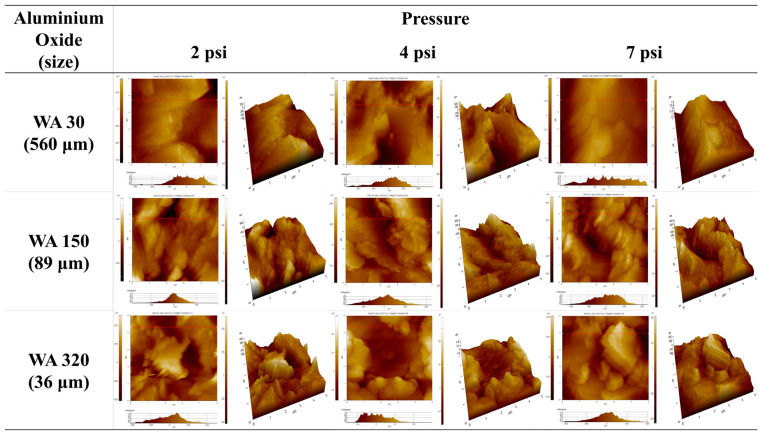
AFM images of SUS treated under various sandblasting conditions before annealing.

**Figure 6 polymers-16-02737-f006:**
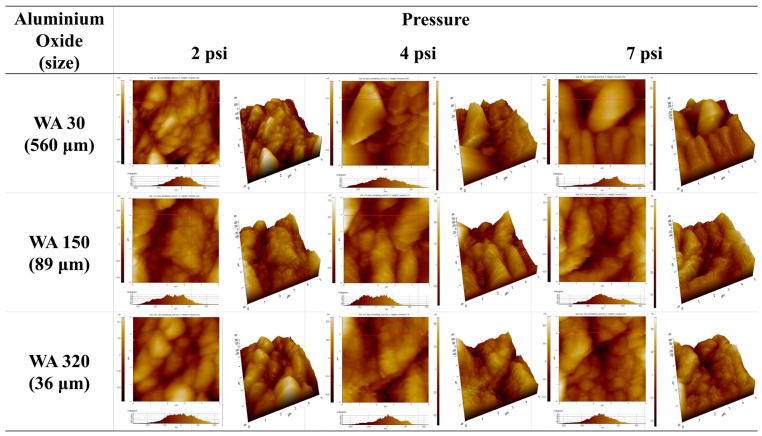
AFM images of SUS treated under various sandblasting conditions after annealing.

**Figure 7 polymers-16-02737-f007:**
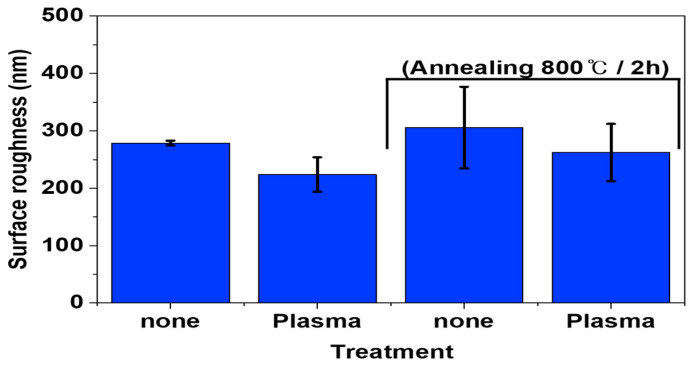
Surface roughness of pristine and plasma-treated SUS/PA66 heterojunction bilayer composites before and after annealing.

**Figure 8 polymers-16-02737-f008:**
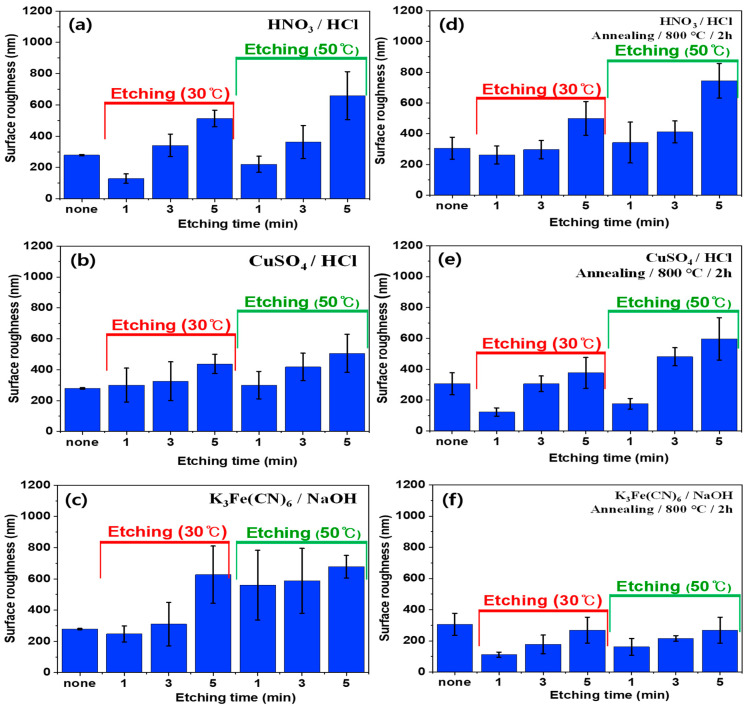
Surface roughness of SUS treated with different etching solutions and conditions before (**a**–**c**) and after (**d**–**f**) annealing: (**a**) HNO_3_/HCl, (**b**) CuSO_4_/HCl, (**c**) K_3_Fe(CN)_6_/NaOH, (**d**) HNO_3_/HCl/annealing, (**e**) CuSO_4_/HCl/annealing, and (**f**) K_3_Fe(CN)_6_/NaOH/annealing.

**Figure 9 polymers-16-02737-f009:**
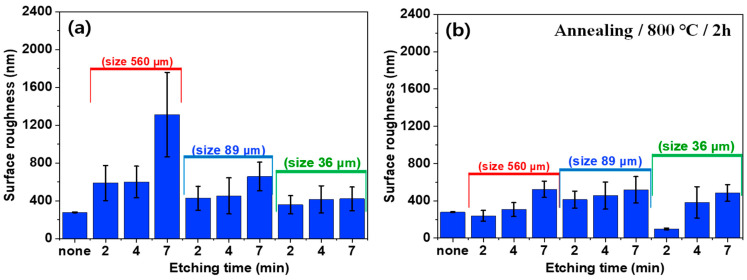
Surface roughness of SUS treated with various blasting conditions before (**a**) and after (**b**) annealing.

**Figure 10 polymers-16-02737-f010:**
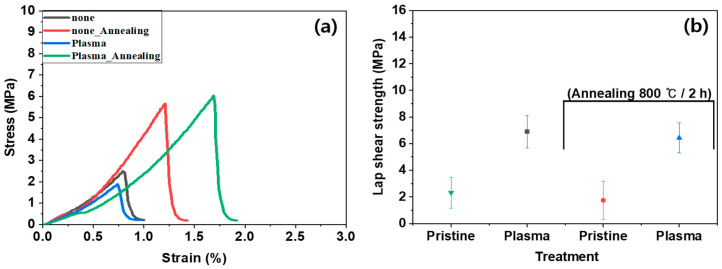
Lap shear strength of pristine and oxygen plasma-treated SUS/PA66 heterojunction bilayer composites before and after annealing: (**a**) stress–strain curve and (**b**) shear strength.

**Figure 11 polymers-16-02737-f011:**
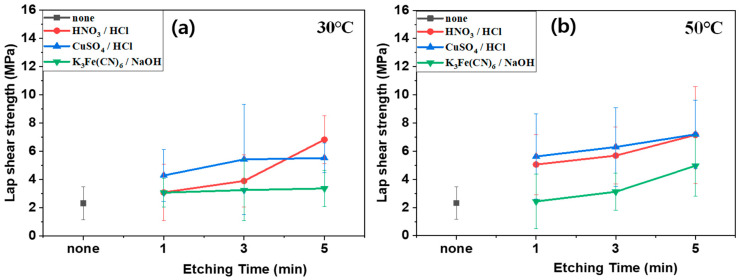
Lap shear strength of heterojunction bilayer SUS/PA66 composites with different etching solutions and conditions before annealing: (**a**) 30 °C and (**b**) 50 °C.

**Figure 12 polymers-16-02737-f012:**
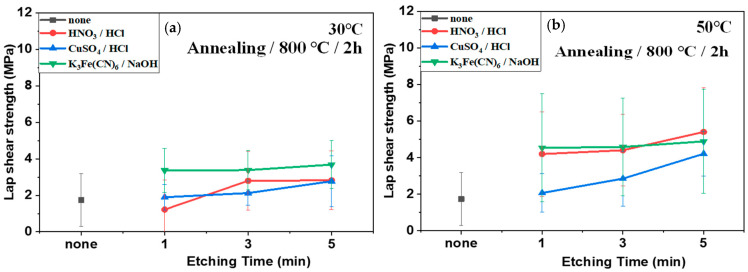
Lap shear strength of heterojunction bilayer SUS/PA66 composites with different etching solutions and conditions after annealing: (**a**) 30 °C and (**b**) 50 °C.

**Figure 13 polymers-16-02737-f013:**
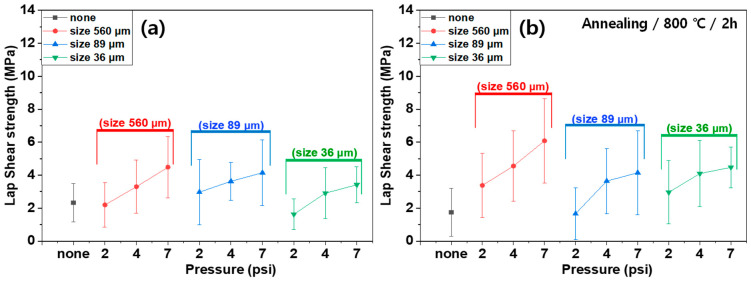
Lap shear strength of heterojunction bilayer SUS/PA66 composites with different blasting abrasive sizes and pressures before (**a**) and after (**b**) annealing.

**Figure 14 polymers-16-02737-f014:**
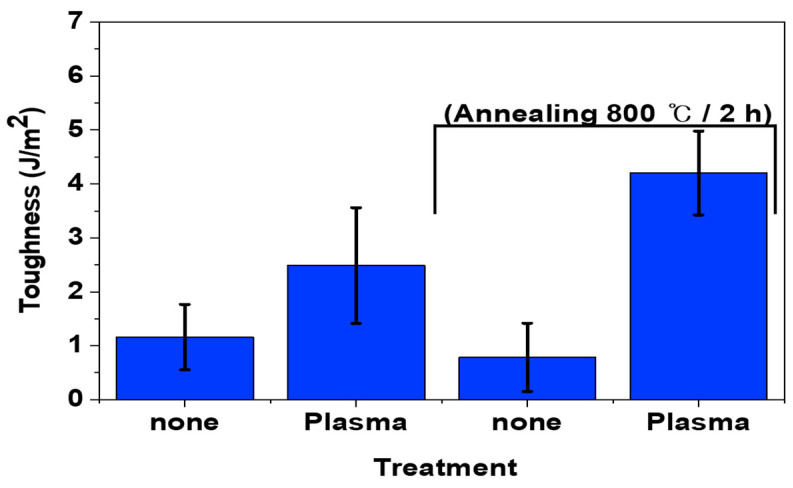
Toughness of pristine and plasma-treated SUS/PA66 heterojunction bilayer composites before and after annealing.

**Figure 15 polymers-16-02737-f015:**
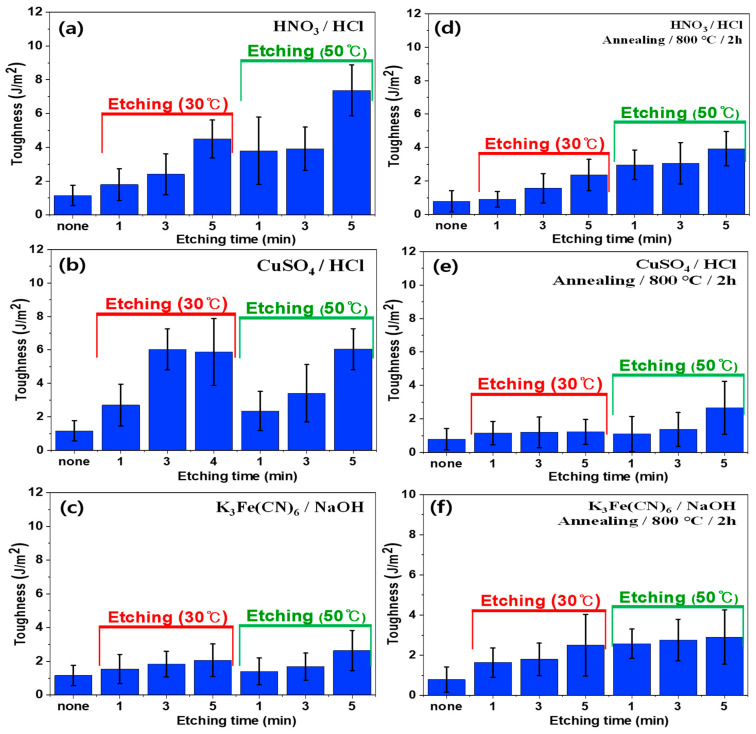
Toughness of heterojunction bilayer composites with different etching conditions before (**a**–**c**) and after (**d**–**f**) annealing: (**a**) HNO_3_/HCl, (**b**) CuSO_4_/HCl, (**c**) K_3_Fe(CN)_6_/NaOH, (**d**) HNO_3_/HCl/annealing, (**e**) CuSO_4_/HCl/annealing, and (**f**) K_3_Fe(CN)_6_/NaOH/annealing.

**Figure 16 polymers-16-02737-f016:**
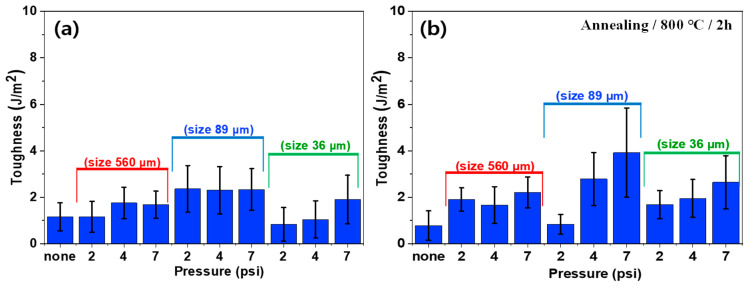
Toughness of heterojunction bilayer composites with different blasting conditions (blasting abrasive size and pressure) before (**a**) and after (**b**) annealing.

**Figure 17 polymers-16-02737-f017:**
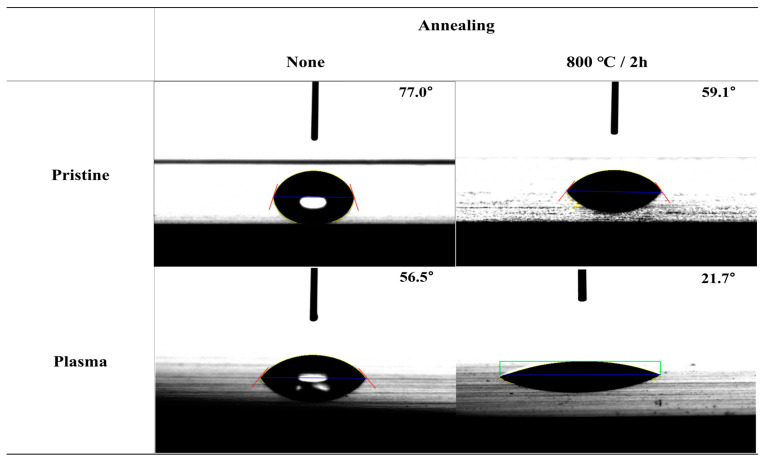
Contact angle of pristine and plasma-treated SUS before and after annealing.

**Figure 18 polymers-16-02737-f018:**
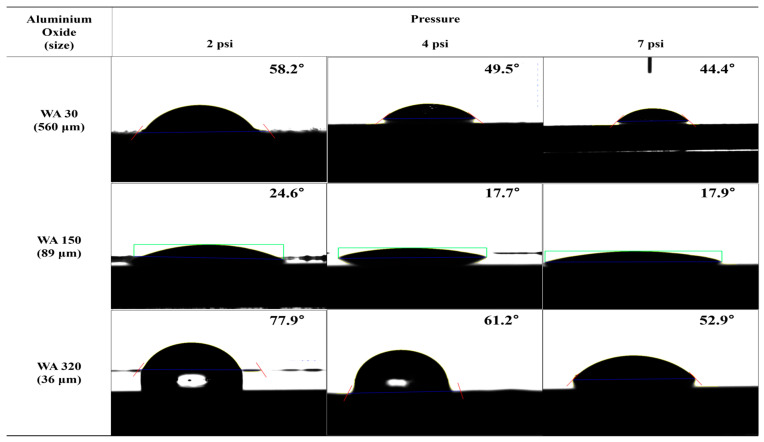
Contact angle of SUS with different blasting conditions before annealing.

**Figure 19 polymers-16-02737-f019:**
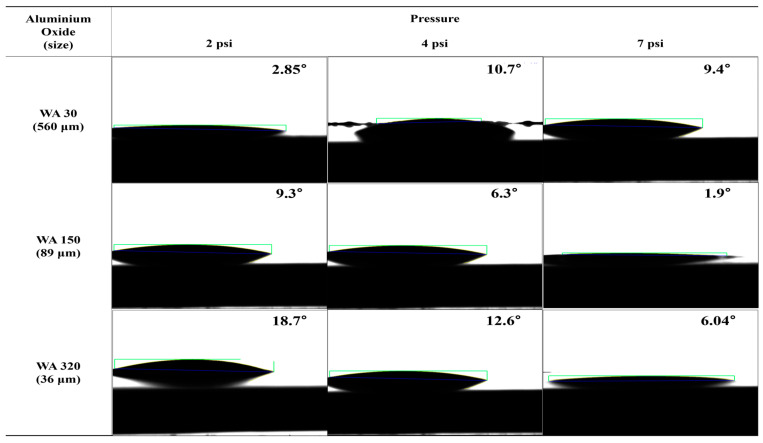
Contact angle of SUS with different blasting conditions after annealing.

**Figure 20 polymers-16-02737-f020:**
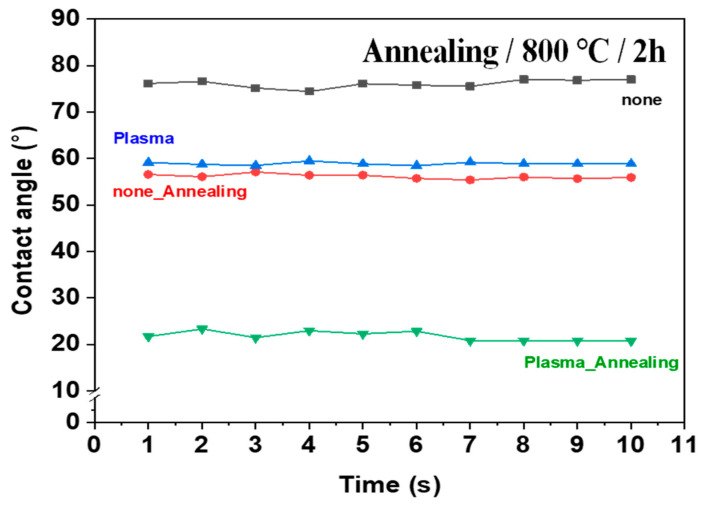
Contact angles of pristine and plasma-treated SUS before and after annealing as a function of time.

**Table 1 polymers-16-02737-t001:** Chemical etching composition and conditions.

Etching Solution (100 mL)	Etching Temperature (°C)	Etching Time (min)
HNO_3_ 20 mL HCl 60 mL DI water 20 mL	30	1
		3
		5
	50	1
		3
		5
CuSO_4_ 5H_2_O 10 g HCl 50 mL DI water 20 mL	30	1
		3
		5
	50	1
		3
		5
K_3_Fe(CN)_6_ 10 g NaOH 10 g DI water 100 mL	30	1
		3
		5
	50	1
		3
		5

**Table 2 polymers-16-02737-t002:** Mechanical etching particle size and pressure.

Aluminum Oxide (Size)	Pressure (psi)		
WA 30 (560 μm)	2	4	7
WA 150 (89 μm)	2	4	7
WA 320 (36 μm)	2	4	7

## Data Availability

The original contributions presented in the study are included in the article/Appendix A, further inquiries can be directed to the corresponding author.

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
