# Peer review of "Optimizing Surface Characteristics of Stainless Steel (SUS) for Enhanced Adhesion in Heterojunction Bilayer SUS/Polyamide 66 Composites"

_polymers, 2024, doi:10.3390/polym16192737_

Round 1

Reviewer 1 Report

Comments and Suggestions for Authors

The review on the manuscript

"Optimizing Surface Characteristics of stainless steel (SUS) for Enhanced Adhesion in Heterojunction Bilayer SUS/Polyamide 66 Composites"

submitted to the journal "polymers"

The manuscript describes the results of the thorough experimental study of surface morphology, mechanical and wetting properties of stainless steel (SUS) covered by polyamide (PA66). The samples were treated in several ways: i) plasma etching; ii) chemical etching (3 reagents, 3 durations and 2 temperatures considered); and mechanical etching (3 powder sizes, 3 pressures). Additionally was considered the effect of annealing at each step, which doubles the amount of the cases. For each case were performedi) SEM and AFM studies which provides information about surface roughness on micrometer scale. ii) stress-strain study, which gives the lap shear strength and toughness of material. iii) wetting contact angle to estimate hydrophilicity. Authors found that different treatments have different effect on mechanical properties of SUS. In particular, the samples treated with K3Fe(CN)6/NaOH improves their shear strength after annealing, while samples treated with HCl based reagents – degrades after annealing. Authors explain such differences by different amount of formed OH groups. The main text and supplementary information presets excessive amount of experimental data with significant efforts to explain them. The manuscript probably could be published, but a few points should be addressed.

Major comments

1. No evidence on the formation of PA66 coating on the samples are given. May be SEM / AFM image of bare SUS samples could be added?

Minor comments

2. The recently published by some of the Authors paper [Polymers 2024, 16, 896. https://doi.org/10.3390/ polym16070896] deals with the selection of compatibizer between the SUS and PA66. However, in reviewed manuscript no compatibizer was mentioned. Why it was not used?

3. The text presentations may be very misleading in the samples nomenclature. The treated objects are mentioned as SUS, which is the stainless steel without coating. The composite materials is mentioned a few times as SUS/PA66. Please clarify if the study performed on bare SUS, or on the composite.

4. Figure S14c (others in lesser degree) show non-monotonous changes, which are not described in the main text. The shear strength of sample treated at 30oC for 1 min is slightly lower than untreated, treated for 2 min is even lower with higher strain, but treated for 3 min – have significantly higher shear strength.

5. Text typography can be improved. Lab or Lap shear strength? Y axis on figures S14, S15, 12a, 13b should be stress, not strength?

6. The twicely repeated statement “Annealing rarely influenced the mechanical properties for the bilayer composites” requires a supporting reference.

Author Response

Please see the attached file with pictures.

Reviewer 2 Report

Comments and Suggestions for Authors

1)    The quantification of the important result analysis is to be presented in the ABSTRACT instead of only qualitative elaboration. 

2)    Introduction is to be still strengthened with the literature review.

3)    The objective and novelty of the research is mentioned at the end of Introduction section.

4)    Please provide the literature support for the etchant composition presented in the Table1. Also, clarify that the etching time mentioned in the Table1 is optimum etching? Please mention in the manuscript.

5)    Lines 183 to 185, it is mentioned as “This enhanced roughness and porosity after annealing were likely to improve the bonding strength of the heterojunction bilayer composites”. Roughness may be enhanced after annealing followed by sand blasting. But how annealing improves porosity? May be reason to be added or a strong literature evidence is to be supported.

6)    Please concise the Conclusions section. Detailed explanation and ‘give reasons’ are not required for this section. Please see that objectives and conclusions are in line.

7)    Please format the Reference section. In some of the paper titles (Ref. no. 3, 9, 13, 14 and 18), the 1st letter of each word is of upper-case letter whereas in other references it is different. Also, in reference 4, the compound name CO2, please correct as CO2 with proper suffix. Please format all the references accordingly.

Comments on the Quality of English Language

Nil

Author Response

(The authors gave the same response as above.)
